# Impact of direct-acting antiviral therapy for hepatitis C–related hepatocellular carcinoma

**Wei-Chen Lin** [1,2,3], **Yang-Sheng Lin**[1,2,3], **Chen-Wang Chang**[1,2,3], **Ching-Wei Chang**[1,2,3], **Tsang-En Wang**[1,2,3], **Horng-Yuan Wang**[1,2,3], **Ming-Jen Chen**[1,2,3]*

**1** Division of Gastroenterology, Department of Internal Medicine, Mackay Memorial Hospital, Taipei, Taiwan, **2** Nursing and Management, MacKay Junior College of Medicine, Taipei, Taiwan, **3** MacKay Medical College, Taipei, Taiwan

* mingjen.4099@mmh.org.tw

**Data Availability Statement:** All relevant data are within the manuscript and its Supporting Information files.

## Abstract

With the introduction of direct-acting antiviral (DAA) agents, hepatitis C virus (HCV) treatment has dramatically improved. However, there are insufficient data on the benefits of DAA therapy in hepatocellular carcinoma (HCC). The purpose of this study was to investigate the outcome of patients who received DAA therapy after HCC treatment. We retrospectively reviewed patients with HCV-related HCC in a single medical center, and the outcome of patients with or without DAA therapy was analyzed. In total, 107 HCC patients were enrolled, of whom 60 had received DAA therapy after treatment for HCC. There were no significant intergroup differences in age, sex, laboratory results, or tumor burden. A more advanced stage was noted in the no DAA group (P = 0.003). In the treatment modality, sorafenib was commonly prescribed in the no DAA group (P = 0.007). The DAA group had a longer overall survival (OS) time than the no DAA group (P<0.001). When stratified by Barcelona Clinic Liver Cancer staging, the DAA group had better OS in the HCC stages 0-A and B-C (P = 0.034 and P = 0.006). There were 35 patients who received DAA therapy after curative HCC therapy. At a median follow-up of 20 months, 37.1% patients had HCC recurrence after DAA therapy. There was no statistical difference in recurrence-free survival between patients receiving and those not receiving DAA (P = 0.278). DAA therapy improved the survival outcome of HCC patients and did not increase recurrent HCC after curative therapy.

## Introduction

Chronic hepatitis C virus (HCV) infection is an inflammatory process of the liver that progressively leads to cirrhosis in about 20–30% of patients [1]. The annual risk of hepatocellular carcinoma (HCC) is around 3% in HCV patients with cirrhosis [1]. Of all HCV-related HCC, 80–90% occur in the setting of cirrhosis [1]. Eradication of HCV has been associated with a decreased risk of developing HCC at any stage of fibrosis [2].

The first anti-HCV drugs, predominantly PEGylated interferon-α and ribavirin, can achieve sustained virologic response (SVR) rates in approximately 55% of patients [3]. The

**Funding:** The author(s) received no specific funding for this work.

**Competing interests:** The authors have declared that no competing interests exist.

innovation of direct-acting antiviral (DAA) agents has dramatically improved the SVR rates to more than 95% in all HCV genotypes and shortened the treatment duration [4]. In 2013, the FDA approved the second generation of DAAs. A recent study showed that DAA therapy also has a safety profile in decompensated liver disease [4]. However, DAA is relatively expensive and, therefore, the cost-effectiveness of treating all HCV individuals, when compared to waiting for treatment until development of a more advanced liver disease or HCC, is controversial. Since January 2017, the DAA was reimbursed by the Taiwan National Health Insurance for HCV patients with advanced hepatic fibrosis or compensated cirrhosis [5]. Since 2019, the health insurance covered all HCV-infected patients and pangenotypic regimens were introduced. There was no prohibition or criteria for using DAA in patients with HCC.

In the interferon era, because fewer cirrhosis patients would tolerate longer duration and side effects, the occurrence of HCC has occasionally been observed after the treatment [6]. In the age of widening use for DAA, initial studies show that this drug has a high risk of short-term recurrence in patients receiving HCC therapy despite viral eradication [6–8]. However, a meta-analysis study shows that there is no evidence that DAA therapy is related to HCC development after adjusting for follow-up period and age [9]. The American Gastroenterological Association (AGA) advises that DAA therapy for HCC patients is available to those who receive curative therapy [10]. In the clinical scenario, a small proportion of Child–Pugh class B cirrhosis patients without curative HCC treatment received DAA therapy for consideration of its benefit of reverse of liver function and then further HCC treatment. Because the continued risk of recurrent HCC persists in patients receiving DAA therapy after curative therapy, the timing and role of DAA therapy are debatable and inconclusive.

Until now, there are few data on the benefit and outcome of HCV eradication in patients who have already developed HCC. Therefore, we conducted this study with the aim of investigating the impact of treatment with or without DAA among different stages of HCC patients undergoing therapy. The second aim was to evaluate the relationship between DAA therapy and recurrent HCC after curative treatment.

## Materials and methods

### Patient selection

This was a retrospective study aimed at assessing the outcome of DAA therapy in HCV-related HCC at the MacKay Memorial Hospital, Taipei. The baseline characteristics and laboratory and radiologic tumor patterns were registered in all patients before starting HCC therapy. The inclusion criteria were as follows: (1) patients with HCV infection having a positive HCV viral load test, (2) patients with HCC diagnosed through pathology or noninvasive imaging, and (3) DAA should be prescribed after the diagnosis of HCC. The exclusion criteria were as follows: (1) concomitant HBV or HIV infection, (2) treatment of HCV with interferon, (3) incomplete HCC follow-up, and (4) terminal HCC.

### Data collection

Altogether, between January 2010 and December 2019, 204 consecutive patients with HCV induced HCC under therapy were registered (Fig 1). Of these, 14 patients with undetectable viral load, and 15 patients with concomitant HBV infection were excluded. During the HCV and HCC treatment, 16 patients with interferon therapy, 15 patients with DAA therapy before HCC diagnosis, 28 patients without complete HCC follow-up, and 9 patients with terminal HCC were not included. Finally, 107 patients were included in this study. Eight patients were diagnosed with HCC before the enrollment period, and the earliest date of diagnosis was December 2004. This study was conducted in accordance with the Declaration of Helsinki and

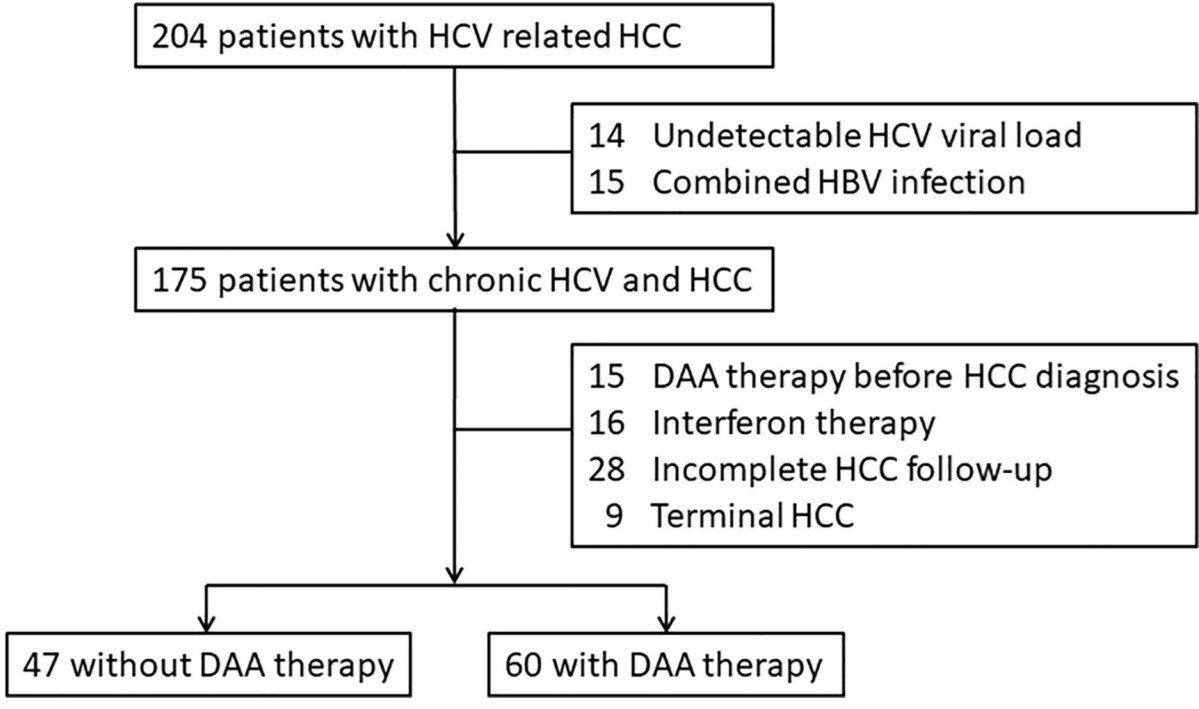

**Fig 1. Flowchart of the patients in the study.**

approved by the Institutional Review Board (IRB) of Mackay Memorial Hospital (19MMHIS284e). This was a retrospective observational study, so the need for informed consent was waived by the IRB. The raw and analysis ready datasets were anonymized by removing all personally identifiable information (**S1 File**).

### Diagnosis and variable definitions

HCC was diagnosed with alpha-fetoprotein (AFP) and imaging studies such as ultrasonography, radiocontrast enhanced triphasic dynamic computed tomography (CT), and/or magnetic resonance imaging (MRI) [11]. The HCC stage was based on the Barcelona Clinic Liver Cancer (BCLC) staging system and classified into five stages (0, A, B, C, and D) [12]. The severity of hepatic fibrosis was assessed by the fibrosis-4 index [13]. Liver cirrhosis was evaluated based on the Child–Pugh classification. Cirrhosis was diagnosed by histopathology or by laboratory tests, ultrasonography, and clinical manifestations of patients with chronic hepatitis with portal hypertension and/or hepatic decompensation [14]. The albumin-bilirubin (ALBI) grade was calculated to predict the prognosis of HCC [15]. The virological response to therapy was assessed by quantitative HCV viral load detection, using real-time polymerase chain reaction with a limit of detection of 15 IU/ml. SVR12 was defined as serum HCV ribonucleic acid levels undetectable 12 weeks after cessation of DAA.

We registered time periods in each patient between HCC diagnosis and therapy, the initiation of DAA, the recurrent HCC after curative therapy, and the last follow-up until December 2019 or HCC-related death. To avoid immortal time bias, all patients who were followed up less than 9 months before DAA therapy were excluded from the analysis. In our hospital, the follow-up policy for HCC patients who achieve complete or partial radiologic response 1 month after therapy is to perform triphasic dynamic imaging. Thereafter, surveillance for HCC consisted of measurements of serum AFP and ultrasonography, CT scan, or MRI scans

of the liver every three to six months [11,16]. Curative therapy was defined as therapy resulting in the disappearance of all target lesions after initial treatment with radiofrequency ablation or hepatectomy. Recurrent HCC was defined as a recently developed nodule with a typical vascular pattern of HCC on dynamic imaging, and its longest diameter is at least 1 cm [17].

## Statistical analysis

Normally and non–normally distributed continuous data were summarized as mean ± standard deviation (SD) and medians with interquartile range (IQR), respectively. Categorical variables were described using frequency distributions and reported as n (%). Student's t-test was used to compare the means of normally distributed continuous variables. The chi-square or Fisher's exact test was used to compare categorical variables. Overall survival (OS) was defined as the interval between the time of HCC diagnosis and death or last follow-up, which was estimated by the Kaplan–Meier method. Recurrence-free survival was defined as the interval between the end of curative therapy and recurrent HCC or last follow-up. The log-rank test was used to determine differences in survival between the groups. Statistical analysis was performed using the STATA statistical package (version 15.0; Stata, College Station, TX, USA). All P values were two-sided, and $P < 0.05$ was considered statistically significant.

## Results

### Baseline characteristics of HCV-HCC patients

Among 107 hepatitis C-related HCC patients, 60 patients were treated with DAA after HCC management (DAA group) and 47 patients did not receive any HCV therapy (no DAA group). The baseline characteristics of the study population are reported in Table 1. The age, sex ratio, and laboratory data of complete blood cell count and liver function were comparable between the two groups. The baseline tumor burden, such as the size of the largest tumor, the number of tumors, and AFP were not statistically different. The proportions of the BCLC stage were 0 (4.3%), A (61.7%), B (17%), and C (17%) in the no DAA group and 0 (31.7%), A (48.3%), B (13.3%), and C (6.7%) in the DAA group. More advanced HCC was noted in the no DAA group (P = 0.003). There were no significant differences in the fibrosis-4 index and ALBI grade between the two groups. In the HCC treatment modalities, targeted therapy of sorafenib was commonly prescribed in the no DAA group (P = 0.007), while other therapies did not differ significantly between groups. In total, 24 patients without DAA therapy and 35 patients with DAA therapy received curative HCC therapy.

### Overall survival in HCC patients with and without DAA therapy

At a median follow-up of 35 months (IQR, 20–56 months), 28% of the patients died. The cumulative incidence of OS at 1, 3, and 5 years was 93.8%, 81.8%, and 61.4%, respectively, in the DAA group and 54.1%, 42.3%, and 24.2%, respectively, in the no DAA group. Patients who received DAA therapy had a significantly longer OS than patients without DAA therapy (P<0.001) (Fig 2A). When stratified by BCLC staging, there was a statistical difference in survival among patients undergoing DAA therapy for the early stage (BCLC stage 0-A) (1-year OS 94.4% vs. 79.6%; 3-year OS, 84.5% vs. 61.9%; P = 0.034) (Fig 2B). In BCLC stage B-C, the DAA group had a significantly higher OS rate than the no DAA group (1-year OS, 91.3% vs. 67.7%; 3-year OS, 65.2% vs. 26.1%; P = 0.006) (Fig 2C).

**Table 1. Baseline characteristics of HCV-related HCC patients.**

| | No DAA N = 47 | DAA N = 60 | P-value |
|---|---|---|---|
| Age, years | 67.2 ± 2.7 | 69.6 ± 2.1 | 0.913[§] |
| Male gender (%) | 25 (53.2) | 31 (51.7) | 0.875[†] |
| Body mass index | 24.5 ± 1.3 | 24.9 ± 1.1 | 0.679[§] |
| HCV viral load, $\log_{10}$ | 6.43 ± 6.11 | 6.36 ± 5.64 | 0.310[§] |
| Albumin, g/dL | 3.5 ± 0.1 | 3.7 ± 0.1 | 0.983[§] |
| Creatinine, mg/dL | 1.03 ± 0.19 | 0.90 ± 0.40 | 0.908[§] |
| Ascites (%) | 4 (8.5) | 4 (6.7) | 0.737[†] |
| Total bilirubin, mg/dL | 1.09 ± 1.05 | 1.08 ± 0.49 | 0.542[§] |
| ALT, IU/L | 72.0 ± 17.8 | 72.7 ± 6.9 | 0.473[§] |
| PT-INR | 1.11 ± 0.01 | 1.12 ± 0.01 | 0.440[§] |
| Sodium, mEq/L | 132.8 ± 4.3 | 139.2 ± 2.5 | 0.103[§] |
| Hemoglobin, g/dL | 12.1 ± 1.7 | 14.3 ± 1.9 | 0.152[§] |
| White blood cell, 10^3/uL | 5.5 ± 2.3 | 4.8 ± 1.8 | 0.958[§] |
| Platelet, k/μL | 122.8 ± 15.8 | 116.5 ± 16.4 | 0.292[§] |
| Fibrosis-4 index | 6.6 ± 1.5 | 7.6 ± 1.5 | 0.167[§] |
| Child-Pugh score (A/B) | 27 / 12 | 47 / 8 | 0.067[†] |
| AFP, ng/mL | 2009.8 ± 1564.2 | 1544.4 ± 1306.9 | 0.591[§] |
| Size of largest tumor (cm) | 4.2 ± 3.3 | 3.0 ± 2.5 | 0.989[§] |
| Tumor no (1/2/3/>3) | 35 / 8 / 1 / 3 | 48 / 6 / 3 / 3 | 0.480[†] |
| BCLC (0/A/B/C) | 2 / 29 / 8 / 8 | 19 / 29 / 8 / 4 | 0.003[†] |
| ALBI grade (1/2/3) | 12 / 31 / 4 | 17 / 42 / 1 | 0.249[†] |
| **Treatment modality** | | | |
| Surgery time | 0.4 ± 0.3 | 0.7 ± 0.5 | 0.763[†] |
| RFA times | 1.2 ± 1.1 | 1.6 ± 1.5 | 0.076[†] |
| TAE/TACE times | 1.2 ± 1.1 | 1.4 ± 0.9 | 0.288[†] |
| PEIT times | 1.7 ± 0.8 | 1.1 ± 0.6 | 0.218[†] |
| Radiotherapy (%) | 4 (7.1) | 1 (1.7) | 0.096[†] |
| Sorafenib (%) | 12 (25.5) | 4 (6.7) | 0.007[†] |
| First line- surgery (%) | 11 (23.4) | 15 (25.0) | 0.849[†] |
| First line- RFA (%) | 15 (31.9) | 28 (46.7) | 0.112[†] |
| Curative therapy (%) | 24 (51.1) | 35 (58.3) | 0.504[†] |

Abbreviations: AFP, alpha-fetoprotein; ALBI, albumin-bilirubin; ALT, alanine aminotransferase; BCLC,Barcelona clinic liver cancer; HCV, hepatitis C virus; PEIT, percutaneous ethanol injection therapy; PT INR, prothrombin time international normalized ratio; RFA, radiofrequency ablation; TACE, Transarterial chemoembolization; TAE, Transarterial embolization

P value was determined using t-test[§] or Chi-squared test[†].

## Characteristics of patients receiving DAA

Among the 60 patients receiving DAA therapy, the most common genotype was genotype 1b (60.0%), followed by genotype 2 (36.7%) (Table 2). Harvoni was the most commonly used DAA regimen in 35% of patients, followed by Sovaldi (16.6%), Zepatier (13.3), and Viekirax + dasabuvir (13.3%). The rapid viral response and SVR12 were 88.3% and 93.3%, respectively. Thirty-five patients (58.3%) received DAA after curative HCC therapy. The median period from curative HCC therapy to DAA prescription was 24 months (IQR, 15–48 months). There were four patients who failed DAA treatment. Among them, three patients (75%) belonged to genotype 2, and three patients (75%) only had a partial response to HCC therapy before DAA

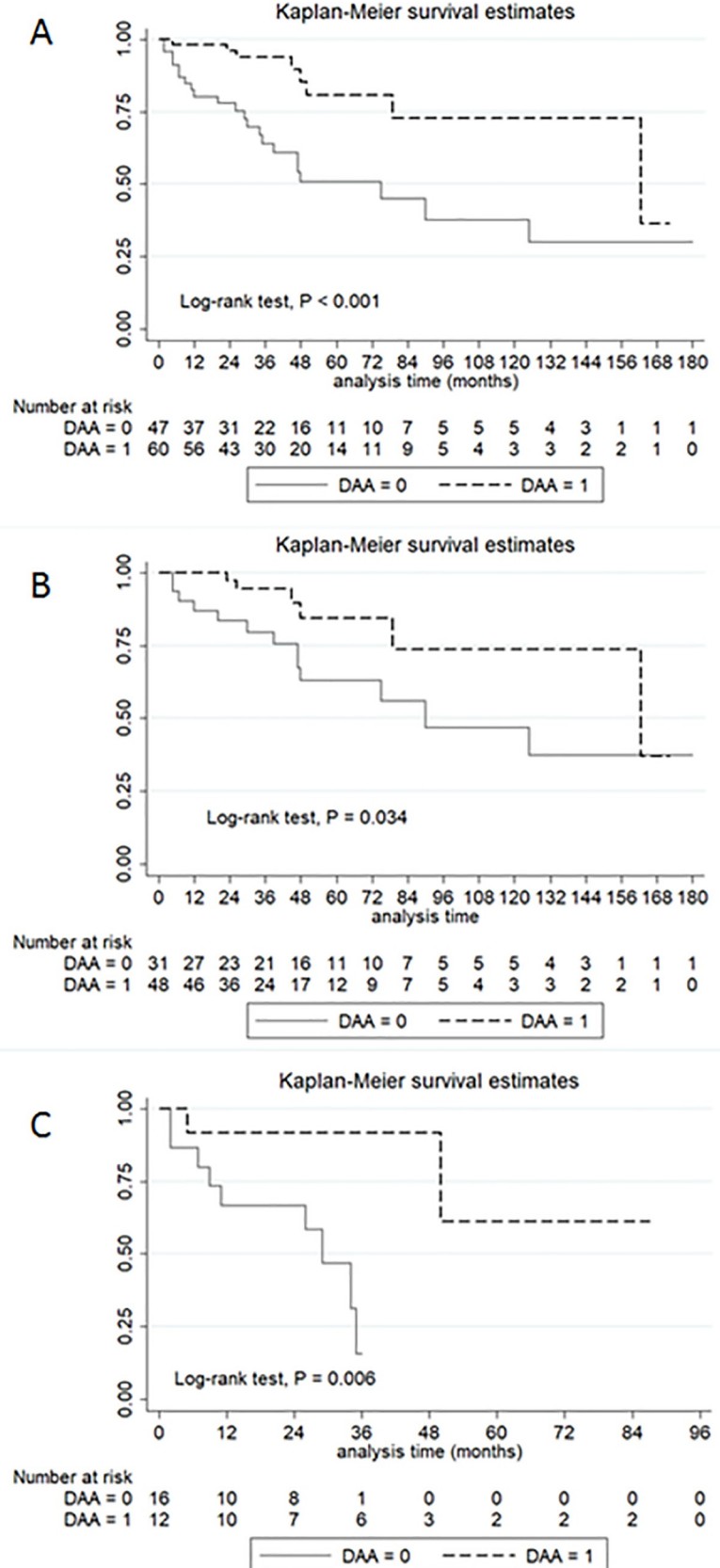

**Fig 2. Overall survival of HCC patients with and without DAA therapy.** Patients with DAA therapy (A), BCLC stage 0-A (B), and BCLC stage B-C (C) had significantly higher overall survival rates than their counterparts.

use. In all, 13 (37.1%) patients had recurrent HCC after curative therapy, and the median follow-up period was 20 months (IQR, 11–26 months).

## Recurrent HCC after curative treatment

Among the 59 HCC patients who received curative therapy, 24 patients (40.7%) had HCC recurrence after the median follow-up time of 21 months (IQR, 12–32 months). The recurrence rates were 37.1% and 45.8% in the DAA and no DAA groups. The cumulative incidences of recurrence-free survival at 1, 2 and 3 years were 85.1%, 73.2% and 40.7%, respectively, in the DAA group and 82.9%, 78.2% and 60.9%, respectively, in the no DAA group. There was no statistically significant difference in the recurrence rate among patients receiving or not receiving DAA (P = 0.278) (Fig 3).

## Discussion

With a higher rate of HCC in patients with HCV than in patients with other causes of cirrhosis [18], realizing the impact of DAA therapy on long-term outcomes in this population is vital. It is likely that there is improved median OS seen in our cohort among patients under HCC treatment receiving DAA therapy. After curative HCC therapy, application of the DAA does not increase or reduce the risk of HCC recurrence in our analysis.

In the pathogenesis of HCV-related HCC, it remains controversial whether the virus plays a direct or indirect role. The core protein of HCV has an oncogenic potential in transgenic mouse models, indicating that HCV is directly involved in hepatocarcinogenesis [19]. In the indirect pathway, HCV causes HCC via chronic inflammation, proliferation, and cirrhosis

**Table 2. Characteristics of the HCC patients who received DAA therapy.**

|  | N = 60 | % or IQR |
|---|---|---|
| **Genotype of HCV** | | |
| 1a | 2 | 3.3 |
| 1b | 36 | 60.0 |
| 2 | 22 | 36.7 |
| **DAA agent** | | |
| Harvoni | 21 | 35.0 |
| Sovaldi | 10 | 16.6 |
| Zepatier | 8 | 13.3 |
| Viekirax + exviera | 8 | 13.3 |
| Sofosbuvir-based | 6 | 10.0 |
| Maviret | 4 | 6.7 |
| Daklinza+ sunvepra | 3 | 5.0 |
| **Rapid viral response** | 53 | 88.3 |
| **12-week sustained viral response** | 56 | 93.3 |
| **DAA therapy after HCC diagnosis (months)** | 25 | 17–42 |
| **DAA therapy after curative therapy (months)** | 24 | 15–48 |
| **Curative therapy (N = 35)** | | |
| Recurrent HCC | 13 | 37.1 |
| Follow-up period after DAA therapy (months) | 20 | 11–26 |

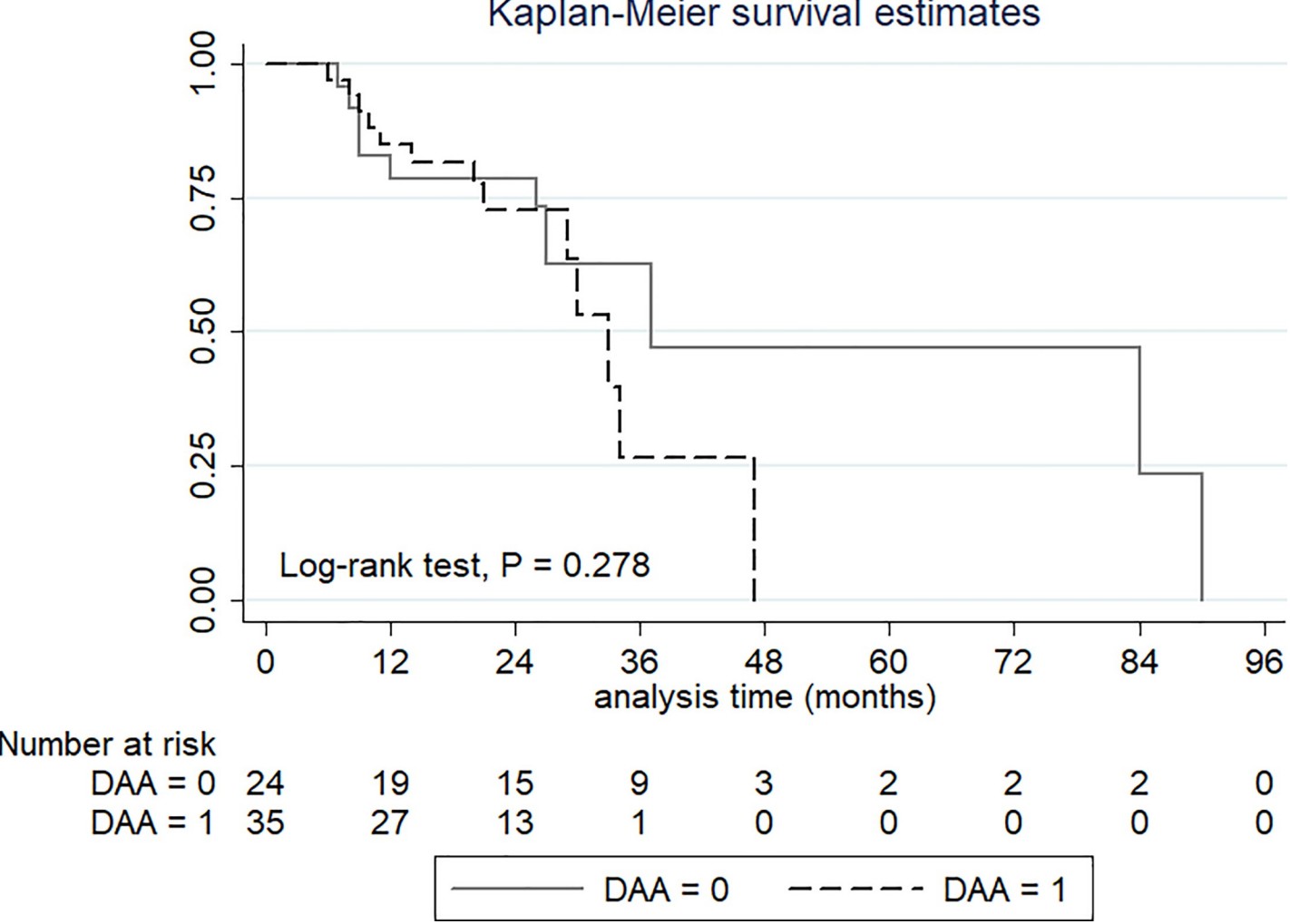

**Fig 3. Recurrence-free survival in HCC patients with and without DAA therapy.**

[20]. The presence of cirrhosis played an important risk factor for developing HCC among HCV patients [18]. HCV genotype 1b–related cirrhosis carry a significantly higher risk of developing HCC than other genotypes [21, 22]. In Taiwan, the Type 1b virus is the predominant genotype of HCV [22], and it was also a major cause of HCC in patients receiving DAA therapy in our study.

There are two factors that determine HCC development [23]. The risk of HCC likely decreases after viral clearance, but the patient's age increases the risk of HCC [23]. Furthermore, patients with a history of HCC present a high risk of recurrent HCC. A large study in Asia showed that the overall occurrence rate of HCC after DAA treatment in patients with previous HCC history was 29.6% and only 1.3% in patients without previous HCC history [6]. The HCC recurrence rate after DAA therapy was higher in this study (37.1%), which may be related to the longer follow-up duration (20 months vs. 15 months) [6]. Due to the risk of HCC recurrence, surveillance at 4-month intervals after DAA therapy in patients with HCC history was recommended [6].

Most patients with HCC are frequently challenged by impaired liver function as a result of cirrhosis, apart from the HCC burden. All guidelines recommend that preserved liver function

is a vital prerequisite to deliver effective HCC treatment [24, 25]. After HCV eradication, liver fibrosis could significantly improve in the long term [26]. One study showed that DAA improves decompensated cirrhosis, which reduces waiting list registrations for liver transplantation [27]. Further, a study proposed a predictive score to evaluate the potential benefits of DAA therapy in decompensated cirrhosis [28]. They found that five baseline factors of body mass index, encephalopathy, ascites, and serum levels of alanine aminotransferase and albumin were related to a reduction of the Child–Pugh score to class A [28]. Therefore, the DAA improved the outcome of HCC, which may relate to the improvement or preservation of liver function. Patients could receive further therapies for the progression or recurrence of HCC after HCV eradication. To consider the risk of liver failure and tumor burden in the terminal stage of HCC, DAA was not prescribed in our general practice; therefore, this stage was excluded. A previous study revealed that the OS in stage D patients was worse in patients adherent to HCC therapy than in nonadherent patients [29]. AASLD guidance recommends that patients with a life expectancy of less than 12 months are unlikely to benefit from HCV eradication and therefore palliative care is suggested [30].

The treatment of HCV-related HCC is controversial since the first published paper showed that recurrent HCC occurred after DAA therapy [31]. However, this study was limited by an increased risk of HCC for patients who did not undergo HCV treatment in comparison to the expected annual risk for patients treated with DAAs. In a large French study and an American cohort study, there does not seem to be an association between curing HCV with DAA agents and an increased risk of HCC recurrence [32]. A review study showed that several risk factors may increase the risk of HCC recurrence after DAA treatment, such as liver cirrhosis, antiviral treatment failure, history of previous HCC recurrence, initial HCC stage, non-curative HCC therapy, and the time interval of DAA initiation [33]. Our study confirmed that there is no evidence for an association between DAA therapy and a higher risk of recurrent HCC after curative therapy. Current experts suggest that it is important to confirm the absence of the tumor prior to starting DAA therapy [23].

Decisions regarding DAA treatment in HCC patients should be considered in light of the tumor stage, liver function, life expectancy, and patient preferences. The optimal timing of HCV treatment in patients with HCC is debatable. Patients who have undergone potentially curative HCC management would not benefit from delayed DAA therapy [9]. The AGA advises that the best timing for DAA therapy is 4–6 months after a complete response to HCC therapy [10]. The variable timing of DAA therapy among HCC patients is noted in this real-world study, and only 58.3% DAA prescription after curative HCC therapy meets the guidelines. In Taiwan, physicians start DAA therapy more quickly after the broadening of the reimbursement criteria and recent studies have shown less evidence of recurrent HCC after DAA therapy [9]. With the sequential application of the available systemic treatment options, especially immunotherapy, an OS of more than two years is feasible for patients with advanced stage HCC [34]. Repetitive locoregional therapies and chronic hepatitis C may affect liver function in the long term. Timing to switch to systemic therapy and HCV eradication to preserve liver function would prolong post-progression survival and improve OS of patients with HCC patients [34,35]. Therefore, individualized DAA treatment among HCV patients in different stages of HCC will enhance the outcome and likely become a future trend.

## Limitations

First, this study was a small, single-center, retrospective design, which might have led to patient selection bias. Few advanced HCC patients underwent DAA therapy since the role of DAA is uncertain, and the possibility of curative therapy is low, resulting in little evidence

about the benefits of DAA therapy in the late stages of HCC. Second, most patients undergo HCV eradication in the era of DAA, which would result in different treatment periods in HCC patients with and without DAA therapy. Third, the timing of DAA therapy may be delayed in some HCC patients due to the different reimbursement criteria, timing of pangenotypic DAA agent introduction, and physician attitude. Large, prospective cohorts with adequately homogeneous patient populations, in terms of HCC staging system, treatment strategy, and evaluation of HCC response after DAA administration, are needed.

## Conclusions

DAA therapy has a survival benefit in patients with HCC. After curative HCC therapy, there is no association between DAA therapy and recurrent HCC.

## Supporting information

**S1 File. Raw data.**
(XLSX)

## Acknowledgments

The authors would like to thank all gastroenterology and hepatology faculty of MacKay Memorial Hospital for excellent clinical assistance and care.

## Author Contributions

**Conceptualization:** Wei-Chen Lin.

**Data curation:** Wei-Chen Lin, Chen-Wang Chang, Ching-Wei Chang, Tsang-En Wang.

**Formal analysis:** Wei-Chen Lin, Yang-Sheng Lin.

**Investigation:** Ching-Wei Chang.

**Methodology:** Yang-Sheng Lin.

**Project administration:** Horng-Yuan Wang, Ming-Jen Chen.

**Resources:** Chen-Wang Chang, Tsang-En Wang.

**Supervision:** Horng-Yuan Wang, Ming-Jen Chen.

**Validation:** Chen-Wang Chang, Tsang-En Wang, Ming-Jen Chen.

**Visualization:** Yang-Sheng Lin, Ching-Wei Chang, Horng-Yuan Wang.

**Writing – original draft:** Wei-Chen Lin.

**Writing – review & editing:** Ming-Jen Chen.

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
