## [Decision Letter · Decision Letter 0]

6 Apr 2020

PONE-D-20-07933

Impact of direct-acting antiviral therapy for hepatitis C–related hepatocellular carcinoma

PLOS ONE

Dear Dr. Chen,

Thank you for submitting your manuscript to PLOS ONE. After careful consideration, we feel that it has merit but does not fully meet PLOS ONE’s publication criteria as it currently stands. Therefore, we invite you to submit a revised version of the manuscript that addresses the points raised during the review process.

We would appreciate receiving your revised manuscript by May 21 2020 11:59PM. To enhance the reproducibility of your results, we recommend that if applicable you deposit your laboratory protocols in protocols.io, where a protocol can be assigned its own identifier (DOI) such that it can be cited independently in the future. For instructions see: http://journals.plos.org/plosone/s/submission-guidelines#loc-laboratory-protocols

We look forward to receiving your revised manuscript.

Kind regards,

Chen-Hua Liu

Academic Editor

PLOS ONE

2. In the ethics statement in the manuscript and in the online submission form, please provide additional information about the patient records used in your retrospective study. Specifically, please ensure that you have discussed whether all data were fully anonymized before you accessed them and the name of the group that waived the requirement for informed consent (whether the IRB or ethics committee, or another entity). If patients provided informed written consent to have data from their medical records used in research, please include this information.

Reviewers' comments:

Reviewer's Responses to Questions

**Comments to the Author**

1. Is the manuscript technically sound, and do the data support the conclusions?

Reviewer #1: Partly

Reviewer #2: Partly

2. Has the statistical analysis been performed appropriately and rigorously? 

Reviewer #1: I Don't Know

Reviewer #2: Yes

3. Have the authors made all data underlying the findings in their manuscript fully available?

Reviewer #1: No

Reviewer #2: Yes

4. Is the manuscript presented in an intelligible fashion and written in standard English?

Reviewer #1: Yes

Reviewer #2: Yes

5. Review Comments to the Author

Reviewer #1: 1. Since the patients who treated with interferon (IFN)-based regimen were excluded in the enrollment per the study design, please correct the following sentence on page 11: “…30 patients received non-DAA therapy (did not receive DAA therapy?)…”.

2. Please make sure that HCV-related HCC BCLC stage 0-A patients who did not receive DAA therapy are 30 instead of 31 as that labeled in Figure 2B.

3. Given the high HCC recurrence rate after curative therapy in this study compared with that reported in reference 6, the authors need to explain the possible mechanisms associated with HCC recurrence.

4. Small case number especially patients in BCLC intermediate or advanced stages is the main shortcoming for this study.

Reviewer #2: This study aimed to investigate the impact of direct-acting antivirals (DAA) on the prognoses of patients with hepatitis C virus (HCV) related hepatocellular carcinoma (HCC). The authors demonstrated that the prescription of DAA after the diagnosis of HCC could achieve an excellent rate of sustained virological response (SVR) (93.3%). Moreover, compared to those without DAA therapy, patients who received DAA had a significantly higher overall survival rate. Of note, among the patients who underwent curative treatments, there was no significant difference in the recurrence rates between patients with and without DAA therapy. Generally, it is interesting and has clinical implications. However, several concerns need to be clarified before drawing this conclusion.

1. In this study, the mean duration from HCC diagnosis to the initiation of DAA therapy was 8.9 months. Moreover, the mean duration from curative HCC therapy to DAA prescription was 11.5 months. It might cause an immortal time bias between patients with and those without DAA therapy after the diagnosis of HCC. There might be two methods to minimize this imbalanced bias. The first one is to perform the date of DAA initiation as a time-dependent covariate. For patients in the DAA group, they were classified as the DAA untreated group before the prescription of DAA (untreated period). They were enrolled as DAA treated group after they received DAA therapy (treated period, from DAA initiation to end of follow-up). The second one is to set a time window. All patients started to be followed up after the index date (maybe 9 months or 1 year after the date of HCC diagnosis). All events occurred before the index date were excluded from the analysis.

2. From the “data collection” in the section of “Materials and methods”, patients were enrolled from January 2010 to December 2019. Moreover, the last follow-up date was December 2019. It seemed that the longest follow-up period was 10 years (120 months). However, in the section of “Results”, the range of follow-up was 1-180 months. Moreover, in the Figure 2A, at least 7 patients (4 in the non-DAA group, and 3 in the DAA group, respectively) had a follow-up duration of 132 months or more. Please clarify it.

3. In the Table 2, among the all 60 patients in the DAA group, 35 patients received HCC curative therapy. However, in the Page 11, Paragraph 2 (recurrent HCC after curative treatment), in the early stage (BCLC stage 0-A) of HCC patients receiving curative therapy, 48 patients received DAA therapy. Please to clarify it.

4. In this study, TACE was regarded as a curative treatment. However, the overall survival rate and tumor control rate were lower in patients who underwent TACE when compared to resection surgery or local ablation therapy. It is suggested to classified TACE as the non-curative treatment modality. Moreover, in the Table 1, it is suggested to show the data of patient number according to their first treatment modality for HCC (for example, how many patients underwent resection surgery at the time of HCC diagnosis between DAA and non-DAA groups).

5. It is suggested to provide the data of FIB-4 and albumin-bilirubin (ALBI) grade in the Table 1.

6. It is suggested to provide the median and IQR when presenting the data regarding the follow-up period.

6. PLOS authors have the option to publish the peer review history of their article (what does this mean?). If published, this will include your full peer review and any attached files.

Reviewer #1: No

Reviewer #2: No

---

## [Author Response · Author response to Decision Letter 0]

16 Apr 2020

Reviewer #1:

1. Since the patients who treated with interferon (IFN)-based regimen were excluded in the enrollment per the study design, please correct the following sentence on page 11: “…30 patients received non-DAA therapy (did not receive DAA therapy?)…”.

Answer 1: Thanks for your comment. We corrected this error. After refining the criteria for curative therapy (initial treatment with surgery or RFA), 24 patients received non-DAA therapy.

2. Please make sure that HCV-related HCC BCLC stage 0-A patients who did not receive DAA therapy are 30 instead of 31 as that labeled in Figure 2B.

Answer 2: Thanks for your comment. There were 31 patients (2 and 29 patients in stage 0 and stage A, respectively) in the non-DAA therapy group, and 30 of them received curative therapy in the initial analysis.

3. Given the high HCC recurrence rate after curative therapy in this study compared with that reported in reference 6, the authors need to explain the possible mechanisms associated with HCC recurrence.

Answer 3: The recurrence of HCC after DAA therapy was higher in this study at 50% (30/60) than in reference 6 (29.6%), since we included patients who showed complete HCC response to TACE therapy and repeated surgery or RFA. After excluding these patients, the recurrence rate in the DAA therapy group was 37.1% (13/35). The follow-up time was longer in this study than in reference 6 (20 months vs. 15 months), which might be another reason. We mentioned this in the discussion. 

4. Small case number especially patients in BCLC intermediate or advanced stages is the main shortcoming for this study.

Answer 4: Thanks for your comment. Because of the small number of HCC patients in advanced stages, this study cannot provide solid evidence of the benefit of DAA in these stages. We mentioned this as the main shortcoming in the discussion.

Reviewer #2: 

1. In this study, the mean duration from HCC diagnosis to the initiation of DAA therapy was 8.9 months. Moreover, the mean duration from curative HCC therapy to DAA prescription was 11.5 months. It might cause an immortal time bias between patients with and those without DAA therapy after the diagnosis of HCC. There might be two methods to minimize this imbalanced bias. The first one is to perform the date of DAA initiation as a time-dependent covariate. For patients in the DAA group, they were classified as the DAA untreated group before the prescription of DAA (untreated period). They were enrolled as DAA treated group after they received DAA therapy (treated period, from DAA initiation to end of follow-up). The second one is to set a time window. All patients started to be followed up after the index date (maybe 9 months or 1 year after the date of HCC diagnosis). All events occurred before the index date were excluded from the analysis.

Answer 1: Thanks for your precious comment. We chose the second method to exclude immortal time bias. After adjustment, the median periods of HCC diagnosis to DAA therapy and curative HCC therapy to DAA prescription were 25 and 24 months, respectively. We revised this in the methods, tables, and results.

2. From the “data collection” in the section of “Materials and methods”, patients were enrolled from January 2010 to December 2019. Moreover, the last follow-up date was December 2019. It seemed that the longest follow-up period was 10 years (120 months). However, in the section of “Results”, the range of follow-up was 1-180 months. Moreover, in the Figure 2A, at least 7 patients (4 in the non-DAA group, and 3 in the DAA group, respectively) had a follow-up duration of 132 months or more. Please clarify it.

Answer 2: Thanks for your comment. Eight patients (5 in the non-DAA group, and 3 in the DAA group) received HCC diagnosis before 2010 and received therapy during the study period of 2010–2019. The earliest date of diagnosis was December 2004. We mentioned this in the methods.

3. In the Table 2, among the all 60 patients in the DAA group, 35 patients received HCC curative therapy. However, in the Page 11, Paragraph 2 (recurrent HCC after curative treatment), in the early stage (BCLC stage 0-A) of HCC patients receiving curative therapy, 48 patients received DAA therapy. Please to clarify it.

Answer 3: Thanks for your comment. Thirteen patients who showed complete response to HCC after TACE or repeated therapy were included. We revised this in the results.

4. In this study, TACE was regarded as a curative treatment. However, the overall survival rate and tumor control rate were lower in patients who underwent TACE when compared to resection surgery or local ablation therapy. It is suggested to classified TACE as the non-curative treatment modality. Moreover, in the Table 1, it is suggested to show the data of patient number according to their first treatment modality for HCC (for example, how many patients underwent resection surgery at the time of HCC diagnosis between DAA and non-DAA groups).

Answer 4: Thanks for your comment. We excluded patients who received TACE therapy as a curative treatment and re-analyzed the data as suggested. We have provided the data on initial treatment modality in Table 1. 

5. It is suggested to provide the data of FIB-4 and albumin-bilirubin (ALBI) grade in the Table 1.

Answer 5: Thanks for your comment. FIB-4 and albumin-bilirubin grade were vital tools to evaluate liver fibrosis and HCC prognosis. We added these data in the table. 

6. It is suggested to provide the median and IQR when presenting the data regarding the follow-up period

Answer 6: Thanks for your comment. We revised all data on the follow-up periods by providing the median and IQR.

---

## [Decision Letter · Decision Letter 1]

27 Apr 2020

PONE-D-20-07933R1

Impact of direct-acting antiviral therapy for hepatitis C–related hepatocellular carcinoma

PLOS ONE

Dear Dr. Chen,

Thank you for submitting your manuscript to PLOS ONE. After careful consideration, we feel that it has merit but does not fully meet PLOS ONE’s publication criteria as it currently stands. Therefore, we invite you to submit a revised version of the manuscript that addresses the points raised during the review process.

We would appreciate receiving your revised manuscript by Jun 11 2020 11:59PM. To enhance the reproducibility of your results, we recommend that if applicable you deposit your laboratory protocols in protocols.io, where a protocol can be assigned its own identifier (DOI) such that it can be cited independently in the future. For instructions see: http://journals.plos.org/plosone/s/submission-guidelines#loc-laboratory-protocols

We look forward to receiving your revised manuscript.

Kind regards,

Chen-Hua Liu

Academic Editor

PLOS ONE

Reviewers' comments:

Reviewer's Responses to Questions

**Comments to the Author**

1. If the authors have adequately addressed your comments raised in a previous round of review and you feel that this manuscript is now acceptable for publication, you may indicate that here to bypass the “Comments to the Author” section, enter your conflict of interest statement in the “Confidential to Editor” section, and submit your "Accept" recommendation.

Reviewer #1: (No Response)

Reviewer #2: All comments have been addressed

2. Is the manuscript technically sound, and do the data support the conclusions?

Reviewer #1: Yes

Reviewer #2: Yes

3. Has the statistical analysis been performed appropriately and rigorously? 

Reviewer #1: Yes

Reviewer #2: Yes

4. Have the authors made all data underlying the findings in their manuscript fully available?

Reviewer #1: Yes

Reviewer #2: Yes

5. Is the manuscript presented in an intelligible fashion and written in standard English?

Reviewer #1: Yes

Reviewer #2: Yes

6. Review Comments to the Author

Reviewer #1: 1. The authors had revised my previous comments point-by-point in its 2nd version.

2. The term “non-DAA therapy” is easily interpreted as “any kinds of antiviral therapies other than DAA” by the readers, I suggest the authors adjust the statement appropriately and consistently in the context.

3. Polish the language by a native English speaker and give the approval certificate to the editorial office.

Reviewer #2: (No Response)

7. PLOS authors have the option to publish the peer review history of their article (what does this mean?). If published, this will include your full peer review and any attached files.

Reviewer #1: No

Reviewer #2: No

---

## [Author Response · Author response to Decision Letter 1]

30 Apr 2020

1.The authors had revised my previous comments point-by-point in its 2nd version.

Answer 1: Thanks for your comment.

2. The term “non-DAA therapy” is easily interpreted as “any kinds of antiviral therapies other than DAA” by the readers, I suggest the authors adjust the statement appropriately and consistently in the context.

Answer 2: Thanks for your comment. We have replaced the term “non-DAA therapy” to “no DAA group” and refined the definition of this group in the results section. 

3. Polish the language by a native English speaker and give the approval certificate to the editorial office.

Answer 3: We have sent the manuscript to an English language editor and provided the associated certificate to the editorial office, as suggested.

---

## [Editor Report · Decision Letter 2]

1 May 2020

Impact of direct-acting antiviral therapy for hepatitis C–related hepatocellular carcinoma

PONE-D-20-07933R2

Dear Dr. Chen,

We are pleased to inform you that your manuscript has been judged scientifically suitable for publication and will be formally accepted for publication once it complies with all outstanding technical requirements.

With kind regards,

Chen-Hua Liu

Academic Editor

PLOS ONE

---

## [Editor Report · Acceptance letter]

12 May 2020

PONE-D-20-07933R2 

Impact of direct-acting antiviral therapy for hepatitis C–related hepatocellular carcinoma 

Dear Dr. Chen:

I am pleased to inform you that your manuscript has been deemed suitable for publication in PLOS ONE. Congratulations! Your manuscript is now with our production department. 

With kind regards,

on behalf of

Dr. Chen-Hua Liu 

Academic Editor

PLOS ONE